# A Preliminary Investigation of *Salmonella* Populations in Indigenous Portuguese Layer Hen Breeds

**DOI:** 10.3390/ani13213389

**Published:** 2023-11-01

**Authors:** Carla Miranda, Sónia Batista, Teresa Letra Mateus, Madalena Vieira-Pinto, Virgínia Ribeiro, Rui Dantas, Nuno V. Brito

**Affiliations:** 11H-TOXRUN–One Health Toxicology Research Unit, University Institute of Health Sciences, Cooperativa de Ensino Superior Politécnico e Universitário, 4585-116 Gandra, Portugal; carla.miranda@iucs.cespu.pt; 2LAQV-REQUIMTE–Associated Laboratory for Green Chemistry, University NOVA of Lisbon, 1099-085 Caparica, Portugal; 3CISAS–Center for Research and Development in Agrifood Systems and Sustainability, Polytechnic Institute of Viana do Castelo, NUTRIR (Technological Center for AgriFood Sustainability), Monte de Prado, 4960-320 Melgaço, Portugal; soniabatista@ipvc.pt (S.B.); tlmateus@esa.ipvc.pt (T.L.M.); mmvpinto@utad.pt (M.V.-P.); 4CECAV–Animal and Veterinary Research Centre, Associate Laboratory for Animal and Veterinary Sciences (AL4AnimalS), University of Trás-os-Montes and Alto Douro, Quinta dos Prados, 5000-801 Vila Real, Portugal; 5EpiUnit–Instituto de Saúde Pública da Universidade do Porto, Laboratório Para a Investigação Integrativa e Translacional em Saúde Populacional (ITR), Universidade do Porto, Rua das Taipas, nº 135, 4050-091 Porto, Portugal; 6AMIBA–Associação dos Criadores de Bovinos de Raça Barrosã, 4730-260 Vila Verde, Portugal; virginia.ribeiro@amiba.pt (V.R.); rui.dantas@amiba.pt (R.D.)

**Keywords:** salmonellosis, One Health, chicken, laying hens, small scale, autochthonous breeds

## Abstract

**Simple Summary:**

Salmonellosis is the second most reported gastrointestinal disorder in the European Union, resulting from the consumption of *Salmonella*-contaminated foods. Chickens are considered reservoirs of this pathogen among food-producing animals, for which hygienic and sanitary measures mitigate the risk to humans through the food chain. However, data about the presence of this pathogen in autochthonous Portuguese chickens or their by-products is scarce. In this context, the aim of this study was to conduct a preliminary investigation on the occurrence of *Salmonella* spp. in autochthonous Portuguese laying hens raised in a semi-extensive system for small-scale production. The screening revealed an absence of *Salmonella* spp. in all cloaca, eggshell, and litter material samples collected (*n* = 279) from the 31 selected flocks. Considering these results and the fact that *Salmonella* is still the leading cause of food-borne outbreaks, the risk posed by Portuguese autochthonous chicken breeds produced through alternative and extensive farming methods can be considered low. However, this risk should not be neglected and needs to be further investigated, using a larger sample size, to validate this trend.

**Abstract:**

The sustainability of agroecological systems, biodiversity protection, animal welfare, and consumer demand for higher quality products from alternative and extensive farming methods have reinforced interest in local breeds that are well adapted to low-input environments. However, food safety needs to be safeguarded to reinforce consumer confidence. The aim of this study was to conduct a preliminary investigation on the occurrence of *Salmonella* spp. in eggshells, hen’s cloaca, and litter materials from autochthonous Portuguese laying hens raised in a semi-extensive system for small-scale production. A total of 279 samples from 31 flocks belonging to 12 farms were obtained, with 63 samples from the “Preta Lusitânica” breed, and 72 samples each from the remaining autochthonous breeds, namely, “Branca”, “Amarela”, and “Pedrês Portuguesa”. None (0%) of the samples analyzed were positive for *Salmonella* spp. To the best of our knowledge, these are the first results of *Salmonella* evaluation from hen’s cloaca, eggshells, and litter materials in autochthonous Portuguese chickens, suggesting that a semi-extensive production system can contribute to better food security and a lower risk to public health and the environment.

## 1. Introduction

Bacterial food infections are a major issue of worldwide public health, and a source of worry for both developed and developing countries. Salmonellosis is the second most commonly reported human gastrointestinal disorder in the European Union (EU), resulting from the consumption of *Salmonella*-contaminated foods [1]. The genus *Salmonella* is characterized as an enteric pathogen affecting mammals, reptiles, and birds and one of the most adaptable environmental pathogens [2]. Clinical signs in humans include gastroenteritis, abdominal cramps, bloody diarrhea, fever, myalgia, headache, nausea, and vomiting [2].

Food-producing animals, particularly chickens, are considered as reservoirs of this agent, which is associated with clinical illness and enormous risk to human health within the food chain [3]. These bacteria cause a high number of food-borne salmonellosis cases annually as a result of eating eggs and raw or undercooked meat contaminated with *Salmonella* [4,5]. However, poor hand washing and contact with infected animals are also some of the contamination routes [6,7]. Salmonellosis is also becoming a major concern associated with ready-to-eat food products not subjected to heat treatment, especially when untreated spices and herbs are contaminated with *Salmonella* [8]. Furthermore, *Salmonella* can spread not only horizontally but also vertically by settling in the reproductive tract of chicken and contaminating fresh eggs [9] and concomitantly chicken embryos may die due to the pathogenicity of *Salmonella* [10]. Therefore, improper treatment of *Salmonella* infections may greatly increase costs for disease management and flock breeding.

The coordinated *Salmonella* control programs implemented by the EU are one of the most celebrated milestones in the fight against zoonotic diseases. The EU established an integrated approach to control *Salmonella* in the food chain involving players at the top government level of the EU Member States, the European Commission, the European Parliament, the European Food Safety Agency, and the European Centre for Disease Prevention and Control [1,11]. These controls and strict measures to reduce the spread of *Salmonella* across the EU require industry-wide proof of its absence as part of buying specifications for raw and finished products. Its absence is proven by the microbiological examination conducted to support both Hazard Analysis and Critical Control Point control and due diligence processes.

The commercial chicken industry has produced highly specialized lines and strains for egg and meat production based on genetic selection for improved performance under controlled breeding conditions [12,13]. *Salmonella* spp. are commensal bacteria frequently present in the intestinal tracts of commercial chickens, but it is unclear if this susceptibility is related or not to the selective breeding for rapid growth and increased feed efficiency.

*Salmonella* infection in chickens involves a complex multistep process, and it has been difficult to pinpoint these mechanisms unequivocally and establish cogent pathways or genetic factors [14]. Several factors such as diet, region of the gastrointestinal (GI) tract, housing, environment, and genetics can influence the microbial composition of an individual bird [15]. Furthermore, the genetic background of the birds, as well as the housing environment, can affect the innate immunity in chickens [16]. Intestinal health issues are very common in high performing poultry lines due to the high feed intake, which puts pressure on the physiology of the digestive system. Excess of nutrients, which are not digested and absorbed in the small intestine, could lead to oxidative stress reactions, impairing the barrier function of the cells lining the gut wall and triggering dysbiosis, i.e., a shift in the microbiota composition in the GI tract, leading to chronic stress and systemic inflammation due to cytokine release [17]. On the contrary, autochthonous chicken breeds are in general birds of slow growth and late maturity, especially when reared in low-input systems. According to Soares et al. [18], Portuguese autochthonous breeds’ growth after 240 days is minimal, despite their comparable growth performances and carcass yields with other European autochthonous chickens raised under similar production systems. Nevertheless, the genetic basis of pathogenesis of *Salmonella* in the chicken host has only been tangentially investigated [14], and further research is needed to determine these genetic factors.

Since indigenous breeds have genetic profiles diverging from those of commercial broilers, it is plausible to hypothesize that some heritage breeds may exhibit a decreased susceptibility to *Salmonella* colonization of the intestine [19].

Biodiversity and the sustainability of agroecological systems are global concerns that are serious, and globally, local varieties and breeds of domesticated plants and animals are disappearing. According to the FAO’s 2019 report [20] on avian species, chickens are the ones with the greatest number of breeds at risk on a global scale. The proportion of avian breeds with unknown risk status is even greater than that of mammalian species, comprising chickens that are a considerable component of the currently extinct species [20,21]. It is estimated that 103 out of the total 1640 chicken breeds identified worldwide have already become extinct, with 95 of them belonging to Europe and the Caucasus, making this region one with the largest number of breeds at risk [20,21]. 

Portugal is a relatively small country with an area of 92,212 km^2^ but with a great variety in its orography and climate conditions, given the diversity of different landscapes leading to a multiplicity of traditional farming systems and several autochthonous animal breeds. Portugal is the European country with the largest number of autochthonous breeds per unit area, four of which are chickens [22,23]. Portuguese chicken breeds like “Pedrês Portuguesa”, “Preta Lusitânica”, “Amarela”, and “Branca” are almost extinct and are currently bred on small-scale farms for egg and meat production for self-consumption, mainly in Northwest Portugal [22,24] within a domestic economy context [25]. Since 2003, conservation programs for local chicken breeds have been developed in cooperation with the breeders’ association (Associação dos Criadores de Bovinos de Raça Barrosã, AMIBA), a genealogical register has been created, and breed standards have been approved.

In recent years, consumers’ knowledge about climate change and their awareness of the impact that intensive animal production systems may have has greatly increased [8]. Furthermore, problems related to biodiversity, competition for land and resources, rusticity, resistance, adaptability, and animal welfare have emerged, strengthening consumers’ concerns about the sustainability of animal production systems [13,25,26,27,28,29,30].

The rediscovery of local products and traditions and renewed consumer interest in products presenting quality traits that are different from those of conventional products have opened the doors to new profitable niche markets [31]. However, consumers’ confidence in the consumption of home-produced eggs, based on control measures applied by health and food authorities [32], should be highlighted due to scarcely available data regarding an increased risk of *Salmonella* infections linked to backyard chicken [33]. 

There are a few studies carried out with Portuguese chicken breeds, mainly very recent and related to phenotypic and productive characteristics, defining patterns and productive systems [25,26,34]. Biometric characterization of the Portuguese hen breeds (“Pedrês Portuguesa”, “Preta Lusitânica”, “Amarela”, and “Branca”) showed a high sexual dimorphism, with the “Branca” breed standing out in all the biometric measures and being better adapted to meat production [25]. The carcass characteristics and meat quality of the “Branca” breed were evaluated by Meira et al. (2022) [35], and they identified an interesting physicochemical profile, with good proportion of minerals, essential fatty acids, and n-3-PUFAs, ensuring that consumers receive a highly nutritional and differentiated product. However, the “Pedrês Portuguesa” and “Amarela” breeds showed a potential for double-purpose production (meat and eggs) [25], with the “Pedrês Portuguesa” standing out as the most productive breed regarding egg production in contrast to the “Preta Lusitânica” with a lower productive capacity [26].

Scarce information exists related to the *Salmonella* infections in chickens of autochthonous Portuguese breeds in extensive or semi-extensive systems. Studies on pathogen agents in local breeds are rare [3,4,19], and there are even fewer works addressing the issue of salmonellosis in Portuguese native chicken breeds [2]. *Salmonella* colonies were not observed in Miranda et al.’s [2] preliminary study, suggesting that autochthonous hen’s eggs produced in semi-extensive systems are not an important vehicle for the infection by *Salmonella*, with a positive impact on animal and public health. 

In light of this knowledge gap and the ubiquity of *Salmonella* in commercial broilers, this study focused on determining the prevalence of *Salmonella* in Portuguese indigenous layer hen breeds. The assurance of a safe and healthy product, produced in extensive or semi-extensive systems, which enhance the sustainability and resilience of the production systems, while adding value to rural economies, is a determining factor for the confidence of consumers in these autochthones chicken breeds, produced locally and in a traditional way.

## 2. Materials and Methods

### 2.1. Sample Size and Distribution

Twelve farms were randomly selected, comprising a total of 558 birds, 497 (89.1%) hens and 61 (10.9%) roosters. They were distributed among the following autochthonous breeds: “Amarela”, with 148 birds ((16 males (M) and 132 females (F)); “Branca”, with 112 birds (13 M and 99 F); “Preta Lusitânica”, with 98 birds (12 M and 86 F); and “Pedrês Portuguesa”, with 200 birds (20 M and 180 F). 

All birds, over the age of 6 months, were listed in the genealogical register of the respective breed and originated from explorations in six different regions of Portugal (Figure 1). These farms are characterized by a small number of birds (less than 50 F) divided into several flocks and usually from different breeds. Each flock has, on average, 1 male for every 10 females. Traditionally, the production of autochthonous chickens has been undertaken for double purposes, egg production (hens), and breeding, fattening, and slaughtering (roosters), with the ideal slaughter weight being achieved in about 9 to 12 months.

Within these 12 hen farms, 31 flocks of chickens were then selected: 7 flocks of the “Preta Lusitânica” breed and 8 flocks of the other breeds. Information regarding the farm’s location, total number of birds raised per farm, chicken type, conditions of bedding, and presence or absence of roosts on the farm was recorded (Table 1). All hen farms included the semi-extensive regime, where the birds spend part of the day outdoors, and most of them have a reduced number of animals (≤50). The farm with the smallest number of birds has 7, while the farm with the largest number of birds has 50 (with an average of 18 birds per farm). 

Concerning medical prophylaxis, all farms administer the mandatory Newcastle Disease vaccine, according to the National Vaccination Plan for Poultry (DGAV–EDITAL No. 3 of Newcastle Disease, 28 March 2019). Furthermore, farms located in Braga and Porto districts have introduced in their programs the Marek disease vaccine. 

### 2.2. Sample Collection

From each hen farm, nine samples were collected during February 2023, including four cloaca samples, four eggshells, and one sample containing litter materials. A total of 279 samples from 31 flocks, belonging to 12 farms, were evaluated in this study, 63 from the “Preta Lusitânica” breed and 72 from each of the remaining autochthonous breeds, namely, “Branca”, “Amarela”, and “Pedrês Portuguesa”. 

Cloaca and eggshell samples were aseptically collected using a sterile swab. The cloaca samples were obtained by the introduction of a swab into the cloacal orifice, and the eggshell samples were obtained by swabbing the entire eggshell surface. Next, each swab was placed inside a sterile tube with 500 µL of buffered peptone water (pre-enrichment in non-selective liquid medium, Scharlau^®^). Litter samples were collected from different zones of the flock using a sterile bag, obtaining one composite and representative sample of approximately 400 g. Samples were refrigerated and transported in a cooling box to the laboratory of microbiology at the IUCS, of the Polytechnic and University Higher Education Cooperative, within 24 h for immediate analysis. 

### 2.3. Isolation of Salmonella *spp.*

The microbiological isolation for the presence or absence of *Salmonella* spp. was performed through the standard method recommended by ISO 6579:2017 [36]. 

Cloaca and eggshell samples were added to 1.5 mL of buffered peptone water and mixed for approximately 2 min. Thirty grams of litter samples were stomached into 120 mL of phosphate-buffered saline for 8 min at 100 rpm, using a Stomacher^®^ (Stomacher^®^ 400 Circulator, Seward Laboratory Systems Inc., Islandia, NY, US) [37].

From each sample, 1 mL was pre-enriched in 9 mL of buffered peptone water and incubated at 37 °C for 18 h. Following the incubation period, 0.1 mL of each sample was inoculated into a modified semi-solid Rappaport-Vassiliadis medium base supplemented with novobiocin (20 mg/L, Liofilchem^®^, S.r.l. Roseto degli Abruzzi, Italy) for selective enrichment at 42 °C for 24–48 h. From the culture obtained using a loopful of colonies, the following selective solid media were inoculated at 37 °C for 24 h: Chromagar Salmonella Plus agar^®^ (CHROMagar^TM^, Paris, France); xylose lysine deoxycholate agar^®^ (Oxoid^®^, Hants, UK), and *Salmonella-Sighella* agar^®^ (Oxoid^®^, Hants, UK). Additionally, the samples were incubated in MacConkey agar (Liofilchem^®^, S.r.l. Roseto degli Abruzzi, Italy) in parallel. Presumptive *Salmonella* colonies isolated were subjected to Gram staining, an oxidase test, and the API 20E identification system (bioMérieux^®^, Marcy l‘Etoile, France) for confirmation, according to the manufacturer’s instructions. *Salmonella* spp. from the collection of the laboratory of microbiology at the IUCS-CESPU were used as controls.

### 2.4. Data Analysis

The data were registered and analyzed using the MS Excel 2016 software. The collected data was registered in a table on an Excel sheet in order to create the different graphs that are presented, as well as to calculate all the percentages.

## 3. Results

### Occurrence of Salmonella *spp.*

The screening of 124 cloaca samples, 124 eggshell samples, and 31 litter material samples showed no presence of *Salmonella.* For each autochthonous breed, all (n = 63) analyzed samples of the “Preta Lusitânica” breed, including 28 cloaca and 28 eggshell samples and 7 litter material samples, tested negative for the presence of *Salmonella*. The same results were observed for the remaining breeds, “Branca”, “Amarela”, and “Pedrês Portuguesa”, each with 32 cloaca and 32 eggshell samples and 8 litter material samples (Figure 2). 

Most samples did not show any growth presumptive of characteristic *Salmonella* colonies on the selective solid media used. These samples were classified as negative, as determined by the observed bacterial growth outcomes and in comparison to the control *Salmonella* strain that was employed. Additionally, a few presumptive colonies were observed, but their identity was not confirmed with the biochemical tests performed in this study. These colonies were also classified as negative (Figure 2). Considering the type of sample, 15% (n = 19) of cloacal, 8% (n = 10) of eggshell, and 16% (n = 5) of litter material samples showed presumptive colonies. In general, the “Preta Lusitânica” breed exhibited more presumptive colonies (n = 7 and 5) than the “Pedrês Portuguesa” breed (n = 3 and 2) for cloacal and eggshell samples from hens, respectively. Among the analyzed litter material, the “Branca” breed exhibited 3 out of 5 presumptive bacterial growth instances, all from different farms. Moreover, a farm that comprised all four breeds showed the majority of presumptive colonies for cloacal (6 out of 19), eggshell (5 out of 10), and litter material samples (1 out of 5).

## 4. Discussion

Food-borne illnesses are an important public health problem worldwide due to the mortality, morbidity, and costs associated with investigations, surveillance, and ultimately the prevention of illness [38]. In Europe, food-borne salmonellosis is the second most commonly reported food-borne gastrointestinal infection in humans among member states, with 60,050 confirmed human cases, 11,785 reported hospitalizations, and 71 deaths in 2021 [11]. Nevertheless, according to the same report, the overall trend for salmonellosis in 2017–2021 did not show any statistically significant increase or decrease. In 2021, 773 outbreaks of salmonellosis were reported, representing 6755 cases, where raw or undercooked eggs and egg-related products were identified as the most important source of these food-borne *Salmonella* outbreaks [11].

Consumer concern regarding the sustainability of production and animal welfare has strongly increased the demand for eggs and meat that are produced through alternative and extensive farming methods [26]. The current shift in consumer preferences for products perceived as “more natural”, “organic”, “humanely-raised”, and viewed as healthier, has led to an increased trend for the consumption of eggs from backyard-raised chickens [32]. Consumer preferences for eggs are mainly driven by intrinsic and extrinsic characteristics, as well as socio-cultural factors [39]. In the end, it is crucial for all stakeholders in the production chain, such as farmers, veterinarians, and stockholders, to work together in close cooperation to fulfill consumer demands for products that are both of excellent quality and safety [40].

While price is very important, especially in developing countries, the production method is nowadays a very relevant factor, from which consumers draw inferences about the health, safety, and sensory properties of eggs. Conventional small-scale egg production, as a source of household food supply, is very popular in the rural areas of Portugal, and frequently, consumers living in urban centers also pursue domestically grown or produced foods. However, little information is available on the conventional small-scale egg production of Portuguese chicken breeds, and to our knowledge, only a few studies were conducted recently [25,26], and only one study was conducted concerning the salmonellosis in Portuguese native chicken breeds [2]. Therefore, the aim of this study was to conduct a preliminary investigation on the occurrence of *Salmonella* spp. in flocks of Portuguese autochthonous hen breeds for conventional small-scale production.

Presenting experimental results regarding autochthonous or indigenous breeds in agriculture or animal husbandry can indeed present some limitations due to the low number of existing farms or populations. Indeed, the small number of farms or birds of a specific autochthonous breed can limit the sample size available for experiments and can result in less statistically robust results. The limited number of chicken farms selected in our study is a reflection of the low number of existing farms with autochthonous chicken breeds in Portugal, since the four autochthonous chicken breeds are at risk of extinction. According to the breeders’ association (AMIBA) data as of 29 August 2023, there were a total of 22,036 Portuguese autochthonous chicken breeds registered (6261 of the “Preta Lusitânica” breed, 3034 of the “Branca”, 5742 of the “Amarela”, and 6999 of the “Pedrês Portuguesa”). However, all these effective breeds are distributed only per 192 of the “Amarela” and “Branca”, 233 of the “Preta Lusitânica”, and 302 of the “Pedrês Portuguesa” breeds, present throughout the national territory and islands, with an average size of 15 to 25 birds per farm (data kindly provided by AMIBA), depicting the abandonment and risk of extinction of these breeds produced under sustainable productive systems. Today, these birds are bred under traditional production systems on small family farms and serve as dual-purpose birds for meat and eggs [18,41]. Females are generally used to produce eggs, while males are kept for meat production and are commonly sold as whole carcasses.

According to our preliminary results, it is plausible to indicate that the products, specially eggs, from Portuguese chicken breeds, produced through alternative and extensive farming methods, could be safe in terms of salmonellosis contamination. The screening for *Salmonella* spp. in a total of 279 samples, including cloaca, eggshell, and litter material, revealed an absence of this bacterium when using a specific growth media. A small fraction of hens within a flock could lead to prolonged opportunities for further horizontal transmission of infection and subsequent egg contamination, and the fecal shedding by infected hens is an important source of *Salmonella* contamination in the chicken housing environment [42]. Effective environmental management of housing systems is essential for minimizing opportunities for the introduction, transmission, and persistence of *Salmonella* in laying flocks.

Chicken litter is a complex material comprised of decomposing plant-based bedding mixed with chicken feces, uric acid, feathers, feed, insects, and other broiler-sourced components, and consequently, the level of pathogens in chicken litter is critical to the overall health of the flock and food safety [43]. Therefore, it is vital to accurately determine if food-borne pathogens are present in litter before, during, and after use.

The other limitation of our study refers to the limited data availability on Portuguese autochthonous breeds with a few baseline data, making it difficult to assess changes or improvements accurately and could hide the establishment of benchmarks and the interpretation of experimental results. Data on *Salmonella* contamination in chicken breeds produced through alternative and extensive farming methods, like backyard eggs, are still very scarce and variable. Some studies have reported the absence of *Salmonella* in backyard eggs analyzed in Spain (n = 10) [44] and Egypt (n = 200) [45], and one study in India showed a 10% (n = 40) occurrence [46]. In Portugal, Ferreira et al. [30] observed that 6 of the 200 eggs sampled were positive for *Salmonella* spp. (3%) and that a positive egg for *Salmonella* spp. was found in 10.7% of the 56 backyard flocks sampled in the north region of Portugal. However, only 1 of the 2 eggs analyzed by Ferreira et al. [32] from each backyard and collected on the same date tested positive, and sampled flocks tested *Salmonella* positive once, i.e., never in both seasons (winter and spring/summer). It is important to highlight that, as in the present study, these previous studies also analyzed only a small number of samples and that makes it impossible to reach reliable conclusions and makes a quantitative comparison with commercial table eggs difficult.

Thus, currently, there is no consensus on which housing systems could influence *Salmonella* contamination. Various factors like housing, temperature, air quality, or light regime may act as stressors, with potentially negative effects on the immune system and consequently less disease resistance [16]. In fact, the particularities of specific management conditions still need to be investigated in more detail.

There is a hypothetical idea that a higher occurrence of *Salmonella* could occur in chicken breeds produced through alternative and extensive farming methods than in commercial methods, considering the absence of preventive measures (e.g., biosecurity programs, vaccination, hygiene practices, and contact with other animals) in these flocks. As an example, egg storage at cold temperatures is a critical factor in preventing *Salmonella* spp. growth in the egg’s content, since feces on egg surfaces increased *Salmonella* spp. growth up to 5 logs during storage at 25 °C [47]. A lack of compliance with safety practices by chicken Portuguese owners was demonstrated by Ferreira et al. [32], as 96% of the eggs were visibly dirty and 92.5% were stored at room temperature.

Previous studies reported that brown eggs have higher quality shells [48,49,50], lower shell permeability [51], and lower penetration ratio of bacteria [52] than white eggs. However, eggshell color cannot be used as a quality assessment tool for nutritive value or safety. According to Messens et al. [53], although brown eggs presented a higher shell thickness and cuticle score, white eggs resisted better *Salmonella* penetration. The authors observed differences in the capacity of eggshells to resist penetration and concluded that these differences cannot be attributed to the genetic strain of the laying hen or housing system. Also, Leleu et al. [54] found a large variation in cuticle coverage and quality within groups of white and brown eggs from old hens, and Ishikawa et al. [55] demonstrated that brown eggshells and their pigments were active against Gram-positive bacteria but not against Gram-negative bacteria including *Salmonella* Enteritidis. However, these results were not corroborated by Dearborn et al. [56], where the proportion of Gram-positive bacteria on eggshells did not vary by egg color.

Genetic and environmental factors regulate hen egg traits. Mori et al.’s [57] results revealed significant effects of breed on eggshell redness and yellowness. According to Lordelo et al. [41], the considerably lighter color of eggshells laid by the Portuguese native breeds, in particular, the “Branca” breed, may be strongly related to their differentiated genetic background, as well as the darker eggshell found in the hybrid breeds, which is probably a consequence of intensive genetic breeding selection. Indeed, several times the brown coloration of the eggshell is a positive influence on consumer preference [49], but the preferences for shell color could also vary worldwide [39]. 

Furthermore, mainly due to concerns about the ethics of the chicken industry and animal welfare, consumers in different countries have shown a marked preference for eggs produced in uncaged systems. Cage-free eggs are often perceived as being of better quality, more nutritious, and safer than caged eggs (reviewed by Rondoni et al. [39]). Lordelo et al.’s [41] results indicated that the overall physical and chemical analyses of the Portuguese native breeds eggs, especially the “Pedrês Portuguesa” and “Preta Lusitânica”, match or supersede the quality of a commercial product in many characteristics. The preference of some consumers for backyard eggs should not be underestimated, leading them to buy eggs from what specialists consider to be uncontrolled sources, for example, eggs that are sold in front of a countryside household after staying for hours at the ambient temperature, in local markets (markets organized for fruits and vegetables where peasants bring eggs), or in front of a food shop where peasants may meet backyard eggs lovers [58]. On the other hand, long-term measures should be implemented to improve food security, reduce the risk to public health [59], and protect the environment. Curiosity aside, certain farms in this study encompassed the four autochthonous breeds, divided into distinct flocks. Nonetheless, most samples that showed presumptive *Salmonella* colonies, including cloaca, eggshell, and litter material samples, originated from a single farm under a semi-extensive regime. Although all samples from the farm tested negative for the presence of *Salmonella*, it is crucial for proper hygienic and sanitary practices to be in place to prevent the dissemination of pathogenic agents like *Salmonella* among flocks. Furthermore, Holt et al. [60] have reported that hens with outdoor access require heightened biosecurity efforts to mitigate potential interactions with predators, wild birds, and rodents. Consequently, such hens face an elevated risk of *Salmonella enterica* infection, leading to *Salmonella*-contaminated egg production [60]. *Salmonella* contamination within hen flocks and eggshells is a multifactorial issue. This contamination is linked to factors such as flock size exceeding >30,000, housing system with high manure contamination levels, significant contamination of egg-handling equipment, and farms with hens of varying ages [61]. Additionally, eggshells are more likely to test positive for *Salmonella* when fecal samples and floor dust samples also yield positive results [61]. This study also emphasizes strategies aimed at reducing these risk factors and effectively controlling *Salmonella* contamination within hen flocks and on eggshells.

## 5. Conclusions

The public health and chicken management problems caused by *Salmonella* infections in laying flocks and contamination of eggs are sufficiently complex that no single control strategy appears likely to provide a completely effective long-term solution. Indeed, it is important to build a strategy upon a foundation of multi-faceted risk reduction practices that include biosecurity, sanitation, pest control, and egg refrigeration. A revision of the current recommendations and regulations is also required, as not all of them ensure that eggs are maintained at temperatures that prevent the growth of *Salmonella* from their collection to their time of purchase. Nevertheless, taking into account these results and the fact that *Salmonella* is still the leading cause of food-borne outbreaks, the risk posed by Portuguese chicken breeds produced through alternative and extensive farming methods can be seen as low. With our preliminary findings obtained from classical methods of isolation of *Salmonella* from the samples, further studies are warranted, including more samples with ample use of traditional and molecular diagnostic tools to confirm our findings.

## Figures and Tables

**Figure 1 animals-13-03389-f001:**
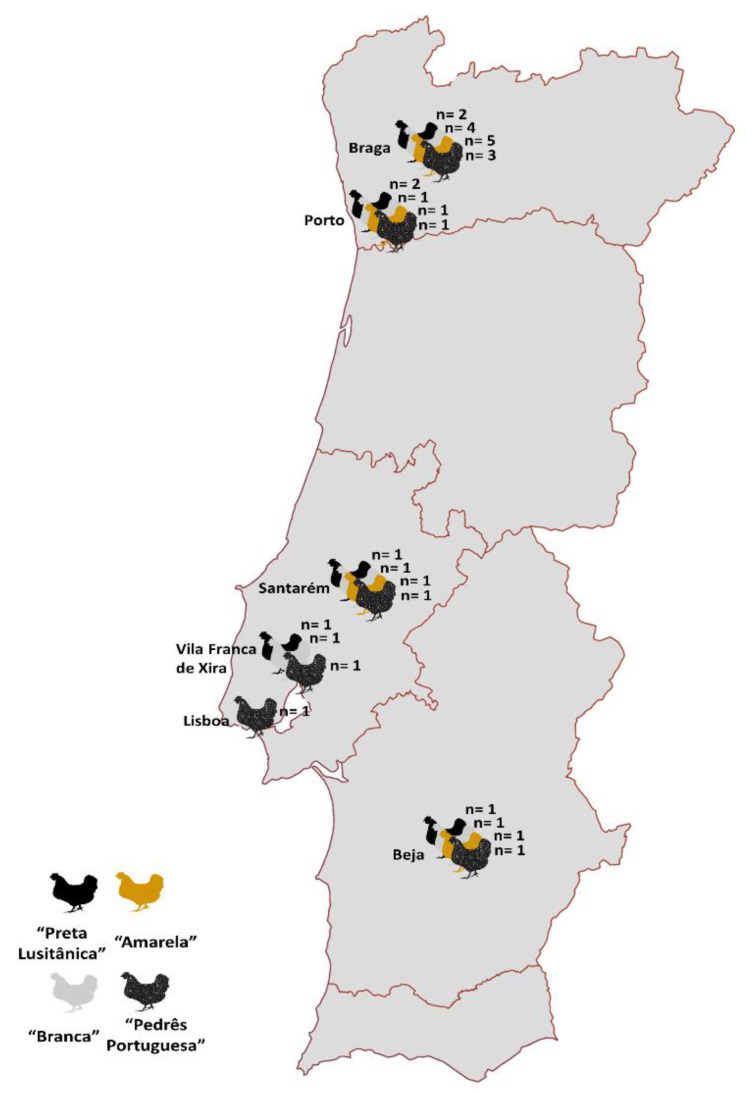
Spatial distribution of the 31 flocks based on the four autochthonous breeds used for this study.

**Figure 2 animals-13-03389-f002:**
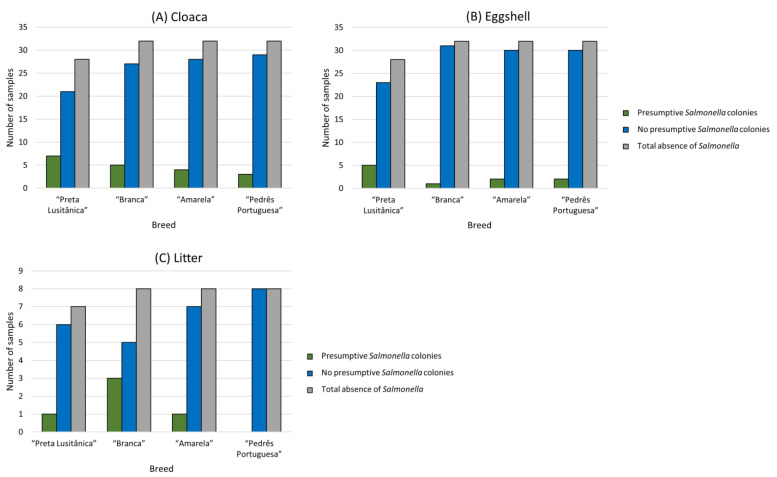
Results of *Salmonella* absence obtained from the bacterial growth, including presumptive colonies or not, in all analyzed samples in this study, comprising the cloaca (**A**), eggshell (**B**), and litter material samples (**C**) per each autochthonous breed.

**Table 1 animals-13-03389-t001:** Characteristics of the hen farms in semi-extensive regime selected for each autochthonous breed like “Preta Lusitânica”, “Branca”, “Amarela”, and “Pedrês Portuguesa” in this study.

Breed/Farm	Region	Total Number of Birds (Male, Female)	Conditions		
			Presence of Roosts	Presence of Bedding	Type of Bedding
“*Preta Lusitânica*”					
1	Porto	16 (2, 14)	Yes.	Yes	Straw
2	Braga	13 (2, 11)	Yes	No	
3	Porto	7 (1, 6)	Yes	Yes	Wood
4	Braga	16 (1, 15)	Yes	No	
5	Santarém	17 (3, 14)	Yes	Yes	Wood
6	Beja	11 (1, 10)	No	Yes	Wood
7	Vila Franca de Xira	18 (2, 16)	Yes	Yes	Straw
“Branca”					
1	Porto	16 (2, 14)	Yes	Yes	Straw
4	Braga	14 (1, 13)	Yes	No	
5	Santarém	11 (2, 9)	Yes	Yes	Wood
6	Beja	10 (1, 9)	No	Yes	Wood
7	Vila Franca de Xira	21 (4, 17)	Yes	Yes	Straw
8	Braga	10 (1, 9)	Yes	No	
9	Braga	14 (1, 13)	Yes	No	
10	Braga	16 (1, 15)	Yes	No	
“*Amarela*”					
1	Porto	17 (2, 15)	Yes	No	
2	Braga	16 (2, 14)	Yes	No	
4	Braga	16 (1, 15)	Yes	No	
5	Santarém	13 (2, 11)	Yes	Yes	Wood
6	Beja	11 (1, 10)	No	Yes	Wood
8	Braga	14 (1, 13)	Yes	No	
9	Braga	15 (1, 14)	Yes	No	
11	Braga	46 (6, 40)	No	Yes	Wood
“*Pedrês Portuguesa*”					
1	Porto	17 (2, 15)	Yes	Yes	Straw
2	Braga	15 (2, 13)	Yes	Yes	Straw
4	Braga	20 (2, 18)	Yes	No	
5	Santarém	46 (6, 40)	Yes	Yes	Wood
6	Beja	11 (1, 10)	No	Yes	Wood
7	Vila Franca de Xira	25 (3, 22)	Yes	Yes	Wood
8	Braga	16 (1, 15)	Yes	No	
12	Lisboa	50 (3, 47)	Yes	Yes	Wood

## Data Availability

Not applicable.

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
