# Peer review of "A Preliminary Investigation of Salmonella Populations in Indigenous Portuguese Layer Hen Breeds"

_animals, 2023, doi:10.3390/ani13213389_

Round 1

Reviewer 1 Report

Comments and Suggestions for Authors

Dear Authors,

Your study aims to investigate the occurrence of Salmonella spp. in eggshells, hen’s cloaca and litter materials from autochthonous Portuguese laying hens raised in a semi-extensive system for small-scale production. In my opinion, it is an interesting case study that should be well-addressed. The MS was well-prepared, but it contained some repeats and broad sentences. Additionally, there are some sentences without references, while others have some nonsense meaning. However, certain aspects may merit publication after some revisions and improvements.

General comments:

Ø Some statements need to be rewritten and others need to be provided with recent references.

Ø Some sections need more details, such as Materials & Methods.

Ø Any scientific name should be written in Italic style.

Specific comments:

L.48: “one health”: What did you mean?

L.52-54: Where is the reference that supports this statement?

L.54-56: Where is the reference?

L.67: After “fresh eggs”, add the following reference:

https://doi.org/10.1155/2022/1393392

L.70: “EU”: Define it.

L.78: “HACCP”: Define it.

L.82-83: “but it is unclear”: Why?

L.84: “non-commercial heritage breeds”: Did you mean “indigenous breeds”?

L.87-89: Remove it (non-sense).

L.89: Add FAO (2019).

L.91-93: Add the missed reference.

L.93-96: Wasn’t clear!

L.108-109: Add a reference supporting your sentences like:

https://doi.org/10.1155/2022/1393392

L.112: “[12, 19-22]”: Add also recent related references like:

- https://doi.org/10.14202/vetworld.2023.369-379

- http://dx.doi.org/10.31893/jabb.22012

 L.135-140: Re-write this paragraph correctly.

L.225: Add the missed necessary details.

L.325-326: Carefully revise these sentences.

Best wishes

Comments on the Quality of English Language

Accepted

Author Response

In attach, the response to the reviewer comments.

Tnak you for the suggestions.

Reviewer 2 Report

Comments and Suggestions for Authors

Miranda et al. performed an investigation on the occurrence of Salmonella spp. in autochthonous Portuguese laying hens raised in a semi-extensive system for small-scale production. The study is very important to understand the epidemiology of Salmonella in Portugal. However, the results are too preliminary to be published. At least the isolated salmonella must be serotyped. 

Comments on the Quality of English Language

Dear Editor, although the manuscript is well-written, the obtained results are too preliminary to be published. At least the isolated salmonella must be serotyped. 

Author Response

Thank you for the comments.

Round 2

Reviewer 1 Report

Comments and Suggestions for Authors

Dear Authors,

After checking the requested revisions and corrections, I am pleased to inform you that your manuscript is now accepted.

Comments on the Quality of English Language

Accept.

Thank you

Author Response

Thanks a lot for your kind suggestions. We have revised the relevant information according to your comments.

Reviewer 2 Report

Comments and Suggestions for Authors

Many thanks to the authors for their response. I propose to to mention the limitation of this study in the discussion. Moreover, some minor comments are in the attached file.

Comments on the Quality of English Language

 Minor editing of English language required

Author Response

Thanks a lot for your kind suggestions. We have revised the relevant information according to your comments.
The appointed  limitation of our study were now mentioned and better discussed between lines 313-331 and lines 348-363.

Specific comments:

L.2: “a” was changed to “A”

L.3: the question mark has been removed

L50. Keywords: “ONE HEALTH” was changed to “One Health”

L90. The paragraph was expanded and it was provided more detailed information based on several bibliographic references, including those suggested by the reviewer.

L91. Word changed to italics.

L162. zeros have been removed

L277. Reference was added.

 The authors appreciate your suggestions